# Are medications safely used by residents in elderly care homes? – A multi-centre observational study from Sri Lanka

**S. M. S. Prasanna‡, T. S. B. Cader‡, S. Sabalingam, L. G. T. Shanika◷, N. R. Samaranayake◍ \*◷**

Department of Pharmacy and Pharmaceutical Sciences, Faculty of Allied Health Sciences, University of Sri Jayewardenepura, Gangodawila, Nugegoda, Sri Lanka

◷ These authors contributed equally to this work.
‡ These authors also contributed equally to this work.
\* nithushi@sjp.ac.lk

**Data Availability Statement:** All relevant data are in OSF: 10.17605/OSF.IO/VJSWA.

**Funding:** The authors received no specific funding for this work.

## Abstract

### Background

Most residents in elderly care homes in Sri Lanka do not receive formal, on-site, patient care services.

### Objective

To evaluate the appropriateness of prescribing, dispensing, administration, and storage practices of medication used by residents in selected elderly care homes in Colombo District, Sri Lanka.

### Methodology

This was a prospective, cross-sectional, multi-center study of 100 residents with chronic, non-communicable diseases, who resided in nine selected elderly care homes in Sri Lanka. Medication histories were obtained from each resident/caregiver and the appropriateness of medications in their current prescription was reviewed using standard treatment guidelines. Prescriptions were cross-checked against respective dispensing labels to identify dispensing errors. Medication administration was directly observed on two separate occasions per resident for accuracy of administration, and matched against the relevant prescription instructions. Medication storage was also observed in terms of exposure to temperature and sunlight, the suitability of container, and adequacy of separation if using multiple medications.

### Results

The mean age of residents was 70±10.5 years and the majority were women (72%). A total of 168 errors out of 446 prescriptions were identified. The mean number of prescribing errors per resident was 1.68±1.23 [median, 2.00 (1.00–3.00)]. Inappropriate dosing frequencies were the highest (37.5%;63/168), followed by missing or inappropriate medications

**Competing interests:** The authors have declared that no competing interests exist.

(31.5%;53/168). The mean number of dispensing errors per resident was 15.9±13.1 [median, 14.0 (6.00–22.75)] with 3.6 dispensing errors per every medication dispensed. Mean administration errors per resident was 0.95±1.5 [median, 0.00 (0.00–1.00)], with medication omissions being the predominant error (50.5%;48/95). Another lapse was incorrect storage of medications (143 storage errors), and included 83 medications not properly separated from each other (58.0%).

## Conclusion

Multiple errors related to prescribing, dispensing, administration, and storage were identified amongst those using medication in elderly care homes. Services of a dedicated consultant pharmacist could improve the quality of medication use in elderly care homes in Sri Lanka.

## Introduction

The proportion of older population is estimated to almost double by the year 2050, and the consequent increasing burden of health of this population is a global concern. Moreover, 80% of this population, is expected to be from low and middle-income countries [1]. The percentage of the older population (60 years and over) in Sri Lanka, a middle-income country has grown dramatically since 1981 [2] and has grown faster than other South Asian countries. In 2012, 1% of the total older population was institutionalized in Sri Lanka [3]. Although caring for this vulnerable group is considered a family obligation by Sri Lankans, a large number of older adults have been institutionalized in the past few decades possibly due to increased youth migration, smaller family size unable to deliver care responsibilities, and the increasing female labor force [4].

Long-term aged care facilities in many countries provide personalized nursing care for residents [5, 6]. In Sri Lanka, however, most patients in these facilities receive medical care from nearby hospital clinics. Most of these facilities do not employ trained healthcare professionals but employ staff who have not received any formal training on safe use of medicines, and a significant proportion are unpaid voluntary workers. Under these circumstances, it is highly likely that prescribing, dispensing and medication administration errors may not be identified by the untrained caregivers.

A study done in the United States (US) found that older adults had the highest age-specific adverse drug event rate; 3.8 per 10,000 persons per year, compared to other age groups [7]. The prevalence is much higher among residents in long-term aged care facilities globally [8–11]. Many studies have also reported a high prevalence of medication errors in long-term aged care facilities compared to hospitals [5, 12]. Published literature pertaining to developed countries report that 16%–90% of residents in these facilities have one or more medication errors [5, 13–15]. As older people experience complex and multiple co-morbidities, they are prescribed numerous medications. Multiple medication use, together with age-related changes in pharmacokinetics and pharmacodynamics, increase vulnerability to adverse drug events [12, 16, 17]. This danger may be augmented by functional disabilities such as visual hearing and mental impairment often seen in older adults.

Although many studies have been conducted on safe use of medications in long-term aged care facilities in high-income countries [5, 13, 14, 18, 19], there is a dearth of such data from low and middle-income countries [20, 21], especially research that assesses the overall

medication process that includes prescribing, dispensing and administration. Clearly, the availability of healthcare services, the quality of care delivery, and health literacy of older people, considerably differ between low and middle-income countries and it is inappropriate to correlate such data from high-and low/middle income countries. Therefore, the aim of this study was to bridge this important information gap on the prevalence of prescribing, dispensing, medication administration, and storage errors in elderly care homes in Sri Lanka.

## Materials and method

### Study design

This study was an observational, prospective, cross-sectional, multi-center study conducted in nine selected elderly care homes in the Colombo District of Sri Lanka.

### Study population

Residents aged 60 years or above with one or more non-communicable chronic diseases (NCCDs) and residing in the selected elderly care home for over three months were recruited for the study. Residents clinically diagnosed with cognitive impairment, those unable to communicate due to functional barriers, and those without competent caregivers to assist, were excluded from the study as they may not have been able to comprehend questions asked by the researchers or provide reliable responses to a medication history interview.

Colombo District was selected for the study through convenience sampling, but it is the District with the highest population density in Sri Lanka. Elders who were 60 years and above, residing in elderly care homes in the Colombo District were eligible for study. According to the list obtained from the Department of Social Services, Sri Lanka, 34 elderly care homes were registered in Colombo District but only 14 were functioning (Fig 1) during the study period. Of these, only nine were selected (366 residents), as officials of the other five elderly care homes did not grant permission for researchers to approach the residents and thereby refused access. Only 100 out of 366 residents in the nine elderly care homes matched the inclusion criteria (Table 1). Therefore, finally, only a convenient sample of 100 older people were selected for the study.

### Study instruments and study process

This study used mixed methods including an interviewer-administered structured questionnaire, review of prescriptions and dispensing labels, and direct observation of medication administration and medication storage practices. Definitions of prescribing errors [22], dispensing errors [23], medication administration errors [22] and storage errors [24] were developed by researchers based on published literature [5] and were used as a guide to identify and classify medication errors (S1 Table).

An in-house pre-tested and validated (content and face) interviewer-administered questionnaire was used to gather information on demographic factors, disease conditions, and medication use practices of residents (S1 File). The research pharmacists obtained a complete medication history from the resident or caregiver and this was followed by a clinical prescription review to identify prescribing errors [25]. The identified problems were recorded in a predefined format. The current and latest available editions (at the time of the study) of British National Formulary (BNF) (Version 71) [26], the Australian Medicines Handbook (AMH) [27] and Medscape Pharmacists [28] were used as references for detecting prescribing errors.

The research pharmacists reviewed medication dispensing labels against the respective prescriptions to assess the appropriateness of the labels [5] and to detect identifiable dispensing

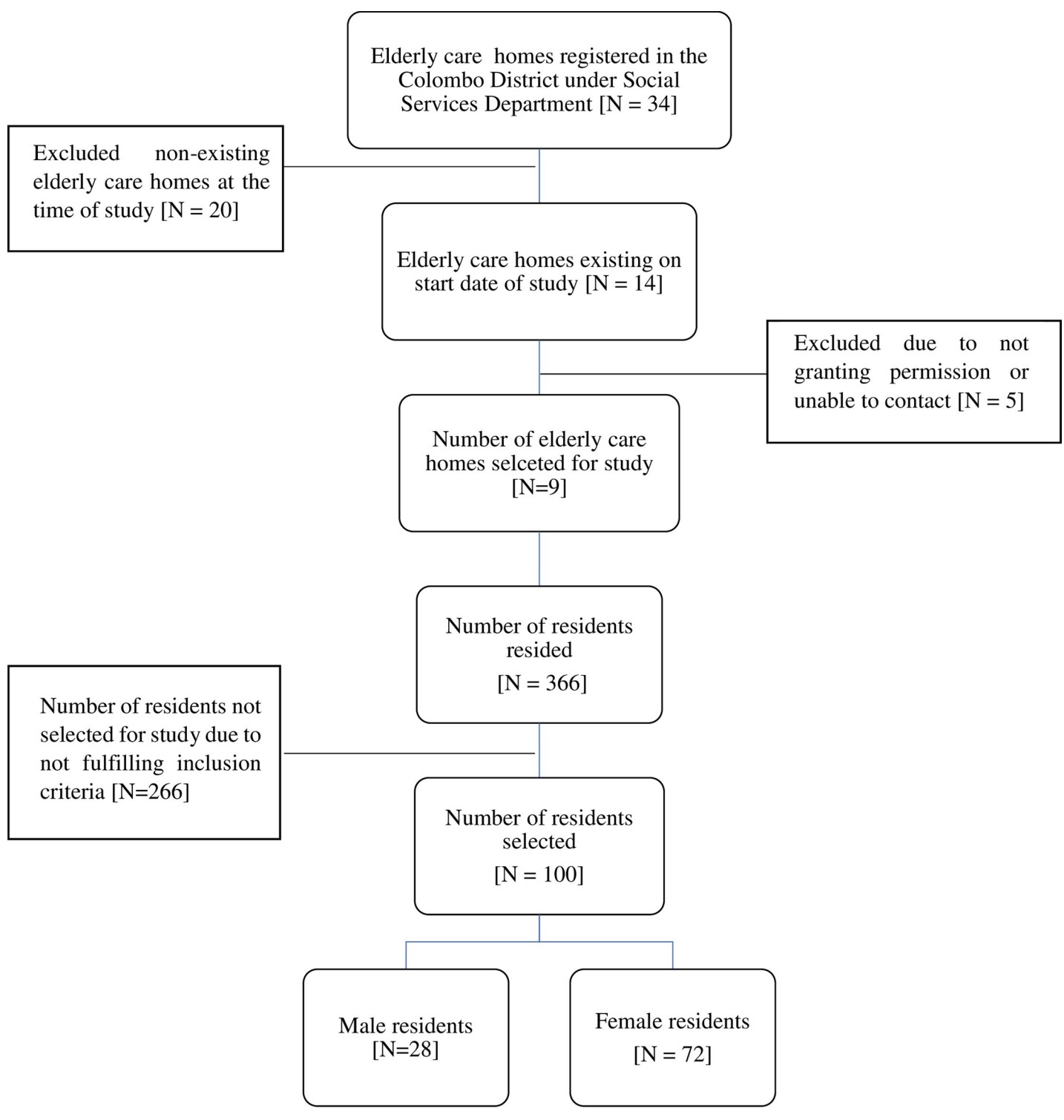

**Fig 1. Flowchart of elderly care homes included in the study.**

errors for each medication. The name, dosage form, strength, dose, frequency, duration, the total quantity of units dispensed, and special instructions were considered in assessing the completeness and appropriateness of medication labels (S2 File). The Australian

**Table 1. Proportion of residents recruited from elderly care homes selected for study.**

| Elderly care home identifier (total number of residents) | Residents selected for study | |
|---|---|---|
| | No of residents (N) | % of all residents living in the care home |
| Elderly care home 01 (N = 85) | 29 | 34% |
| Elderly care home 02 (N = 68) | 11 | 16% |
| Elderly care home 03 (N = 15) | 2 | 13% |
| Elderly care home 04 (N = 15) | 5 | 33% |
| Elderly care home 05 (N = 16) | 6 | 38% |
| Elderly care home 06 (N = 50) | 20 | 40% |
| Elderly care home 07 (N = 56) | 6 | 11% |
| Elderly care home 08 (N = 49) | 16 | 33% |
| Elderly care home 09 (N = 12) | 5 | 42% |
| Total | 100 | 100 |

Pharmaceutical Formulary and Handbook (APF) (Version 21) [29] was used as a reference to assess the appropriateness of medication dispensing. This is an accepted reference used to guide pharmacists on good dispensing practices and contains the relevant instructions for dispensing labels.

The research pharmacists observed two medication administration cycles per resident: one morning cycle and one evening or night cycle, each on a separate day to identify administration errors [5, 30]. The two research pharmacists observed if the right medication was administered in the right dose (strength x number of medication units), at the right time and by the correct route against the most recent prescription of the resident. Any administration discrepancies were recorded as an administration error (S3 File).

Storage practices were directly observed by the research pharmacist [31] using a pre-defined checklist (S4 File) based on product information leaflets of relevant medications and referring to the Australian Pharmaceutical Formulary and Handbook (APF 21) [29].

Data were collected during a period of three months (from 1st December 2016 to 28th February 2017). The questionnaires and checklists used in the study were developed and reviewed by two experienced academic pharmacists. The questionnaire and checklists were then content validated by two other academic pharmacists and were piloted for feasibility in 10 older adults who were not part of the study. Data cleaning and an external audit was undertaken by an independent investigator. All problems related to prescribing, dispensing, medication administration, and storage that were identified during the study were checked and endorsed by two academic pharmacists and 100% consensus was achieved by discussion and re-categorizing where necessary.

## Data analysis

Data were analyzed using SPSS, Version 16. Descriptive statistics are shown as means and frequencies with standard deviations. Prescribing errors were retrospectively categorized using part of the National Coordinating Council for Medication Error Reporting and Prevention (NCC MERP) Index for Categorizing Medication Errors shown in S2 Table [32].

## Ethical consideration

Ethical approval was obtained from the Ethics Review Committee, Faculty of Medical Sciences, University of Sri Jayewardenepura, Sri Lanka (B. Pharm 01/2016). Permission was obtained from the Social Services Department in Sri Lanka which is the authorized regulatory body for

the management of elderly care homes and from the manager/head of each elderly care home prior to the study. Written and verbal consent in the resident's own language was obtained from all the residents who participated in the study. The purpose of the trial, the voluntary nature of the consent and the ability for participants to withdraw the consent were clearly explained before obtaining consent.

## Results

The study included 100 residents in nine elderly care homes in the Colombo District. The mean age of residents was 70±10.5 years and the majority were women (72%) (Table 2). The most prevalent NCCD was hypertension (30.7%) followed by diabetes mellitus (27.8%) (Table 2).

A total of 99 residents had at least one medication error (prescribing, dispensing, administration or storage errors).

## Prescribing errors

Among 446 medications reviewed, 168 prescribing errors were identified. The mean number of prescribing errors per resident was 1.68±1.23 [median, 2.00 (1.00–3.00)]. Eighty-five percent of residents had at least one prescribing error.

**Table 2. Demographics of the study population.**

| Demographics | Outcome |
|---|---|
| **Age (in years), mean (SD)** | 70 (10.5) |
| **Gender, %** | |
| Men | 28 |
| Women | 72 |
| **Highest Level of Education, %** | |
| No schooling | 52 |
| Grade 1–5 | 20 |
| Grade 6–11 | 15 |
| Completed Ordinary Level Examination | 7 |
| Completed Advanced Level Examination | 6 |
| **Assistance in medication administration, %** | |
| By self | 34 |
| By caregiver | 66 |
| **Diagnosis, %** | |
| Hypertension | 30.7 |
| Diabetes mellitus | 27.8 |
| Dyslipidemia | 12.7 |
| Mental health problems | 6.9 |
| Bronchial asthma | 4.6 |
| Gastroesophageal reflux disease | 2.3 |
| Myocardial infarction | 2.3 |
| Others[a] | 12.7 |
| **Number of medications per resident, Mean (SD)** | 4.46 (2.3) |

SD = Standard deviation

[a] Heart failure, hyper/hypothyroidism, osteoarthritis, rheumatoid arthritis, epilepsy, cerebral atrophy, parkinsonism, glaucoma

The most common types of prescribing errors were prescribing incorrect frequency accounting for 37.5% of the total, followed by prescribing wrong medications accounting for 31.5% (Table 3). Losartan (35%; 22/63) and omeprazole (21%; 13/63) were the medications most associated with incorrect frequencies as they were prescribed two times a day, deviating from the recommended guidelines.

Wrong medications were further classified into four sub-categories (unnecessary medications, duplication of medications and inappropriate medication combinations, drug interactions, and potential adverse drug reactions (ADRs)). Table 3 shows the prevalence of sub-categories of wrong medications.

In total, nine drug interactions were identified (Table 3), of which two were considered serious. These interactions should have been addressed by using alternative medications. Seven potential adverse drug reactions were identified which required close monitoring of the residents. None of these residents appeared harmed at the time of the study and hence belonging to NCC MERP Category C (an error occurred that reached the patient but did not cause patient harm) or Category D (an error occurred that reached the patient and required monitoring to confirm that it resulted in no harm to the patient and/or required intervention to preclude harm). Residents were not monitored beyond the point of data collection; hence we could not conclude if these prescribing errors resulted in hospitalization or mortality at a future date.

## Dispensing errors

Mean dispensing errors per resident was 15.9±13.1 [median, 14.0 (6.00–22.75)], and accounted for 3.6 dispensing errors for every medication dispensed. Ninety-five percent of residents had medications with at least one dispensing error. Only 2.4% (11/446) of medications had complete dispensing labels.

Absence of information on the duration of treatment in the dispensing label was the commonest dispensing error (19.5%), followed by lack of provision of medication specific special instructions (18.6%) (Table 4). More than 75% of the dispensing labels contained only dose and frequency.

## Medication administration and storage errors

Mean medication administration errors per resident were 0.95±1.5 [median, 0.00 (0.00–1.00)]. Ninety-five medication administration errors were identified and there were 45 residents with at least one administration error. Twenty-seven of these residents were assisted by caregivers and 18 administered medications themselves. One half (50.5%) of the medication administration errors were due to medication omissions, followed by wrong dose (23.2%), wrong timing (15.8%), extra doses (7.3%), and wrong frequency (3.2%) (Table 5). Wrong drug, wrong dosage form, and wrong route of administration errors were not observed.

Among the medications evaluated, there were 143 storage errors, of which 83 (58%) medications were not properly separated from other medications when storing; 50 medications were stored in containers contrary to recommendations (ex: Storing in plastic bags, sheet of paper to wrap the tablets), and 10 were exposed to inappropriate temperatures and excess sunlight (Table 6).

## Discussion

The logistical steps in the medication use process in general comprise prescribing, dispensing, medication administration and monitoring of medication use [33]. The latter processes should entail the care and oversight of several healthcare professionals in order to ensure safe delivery

**Table 3. Types of prescribing errors.**

| Type of prescribing errors | Frequency (%) [N = 168] * | Examples |
|---|---|---|
| Wrong frequency | 63 (37.5) | • Nifedipine 20 mg immediate-release tablet was prescribed once a day for hypertension instead of three times a day. |
| | | • Betahistine 16 mg was prescribed once a day for vertigo instead of three divided doses. |
| | | • Long-acting glibenclamide 5 mg three times a day was prescribed instead of once a day. |
| Wrong medication | 53 (31.5) | • Glibenclamide 5 mg three times a day was prescribed which could induce hypoglycemia in older adults due to its long action. |
| *I. Potential adverse drug reactions* | *7 (4.2)* | • Prazosin 2 mg three times a day was prescribed as an antihypertensive which increases the risk of hypotension in older adults. |
| *II. Unnecessary medications* | *29 (17.3)* | • Vitamins, calcium supplements, proton pump inhibitors, antihistamines, and non-steroidal anti-inflammatory medicines prescribed without evidence of a justifiable indication. |
| *30 Drug interactions* | *9 (5.3)* | • Metoprolol and amlodipine prescribed together which could cause a significant drug interaction by increasing anti-hypertensive effect. Advised to monitor closely. |
| | | • Olanzapine and benzhexol prescribed together which could cause a significant drug interaction by increasing anti-cholinergic side effects. Advised to monitor closely. |
| | | • Fluoxetine and haloperidol prescribed together which could cause a serious drug interaction. Fluoxetine inhibits the metabolism of haloperidol by inhibiting metabolism by hepatic enzyme CYP2D6. Advised to use an alternate medication. |
| *10 Duplications or inappropriate medication combinations* | *8 (4.7)* | • Prescribed both losartan and enalapril together which is a dual blockade of the renin-angiotensin system. |
| | | • Prescribed both omeprazole and famotidine together which are acid-suppressing agents when either one is adequate. |
| | | • Prescribed two types of vitamin B supplements to the same resident. |
| | | • Prescribed both salbutamol oral tablets and metered-dose inhaler when inhaler alone is adequate and safer in asthma management. |
| Omission of medications | 25 (14.9) | • Not prescribing of a proton pump inhibitor for a resident who was on both aspirin 75 mg and clopidogrel 75 mg. |
| Wrong dose | 16 (9.5) | • Omeprazole 40 mg twice a day prescribed to prevent non-steroidal anti-inflammatory drug (NSAIDs) associated gastric ulcers, but 20 mg daily is the preferred dose. |
| | | • Spironolactone 20 mg mane was prescribed instead of 25 mg. |
| Untreated indications | 9 (5.4) | • Resident's past clinic records indicate hypertension (repeatedly high blood pressure values) but no antihypertensive prescribed (excluded intentional omission). |
| Wrong dosage form | 2 (1.2) | • Residents with a past history of non-adherence to medications was prescribed diltiazem 30 mg three times a day which could be converted to once a day sustained-release form to improve adherence. |
| | | • Salbutamol tablets prescribed instead of a salbutamol inhaler for a resident with chronic bronchial asthma |

*Used as the denominator for calculating percentages

of medications. It is evident from our study that residents evaluated in nine elderly care homes in Sri Lanka were potentially susceptible to medication errors by virtue of various lapses and lack of oversight. Alarmingly, almost all (99%) of the elderly care home residents who participated in our study were exposed to one or more medication errors. Although not comparable with our study, Hanlon et al., [34] too showed that more than 90% of elderly inpatients in United States received one or more inappropriate medicines with high-severity outcomes [34].

There was a relatively high prevalence of one or more prescribing errors amongst some three quarters (85%) of the evaluated cohort. However, it is difficult to compare this result with most other data where disparate instruments have been used to assess inappropriate medication use, for instance Barber N et al. [5], Rousseau A et al. [19], Storms H et al. [35] used Beers' criteria, Screening Tool of Older People's Prescriptions (STOPP), Screening Tool to Alert to Right Treatment (START) criteria, and Medication Appropriateness Index (MAI).

**Table 4. Types of dispensing errors.**

| Types of dispensing errors | Frequency, (%) [N = 1594]* | Examples |
|---|---|---|
| Wrong or missing duration of treatment on a dispensing label | 311 (19.5) | Domperidone duration was missing in the dispensing label. |
| Wrong or missing special directions for use | 296 (18.6) | Twice a day dose of furosemide was written as morning and night in the dispensing label instead of morning and 2.00 p.m. |
| Wrong or missing name of medication on a dispensing label | 280 (17.6) | The medication name was missing on the dispensing label when dispensing carbamazepine. |
| Wrong or missing dosage form of medication on a dispensing label | 235 (14.7) | Nifedipine 20 mg sustained release form was indicated on the dispensing label of a medication pack containing nifedipine 20 mg immediate-release form. |
| Wrong or missing dose of medication on a dispensing label | 231 (14.5) | Dose of losartan potassium was missing on the dispensing label. |
| Wrong or missing the total number of medication units dispensed on a dispensing label | 135 (8.5) | Total number of folic acid tablet units dispensed was missing on the dispensing label. |
| Wrong or missing frequency of medication on a dispensing label | 106 (6.6) | Frequency of metformin administration was missing on the dispensing label. |

*Used as the denominator for calculating percentages

Although there are no published data on elderly care homes, two previous hospital studies conducted in Sri Lanka have identified that nearly 70% of non-communicable chronic disease patients had inappropriate medications in their discharge prescriptions [36, 37]. The types of prescribing errors frequently observed in our study is also consistent with another study conducted among in-patients in a tertiary care hospital in Sri Lanka [22]. According to two recent systematic reviews published by Storms et al. 2017 [35] and Ferrah et al. 2016 [14], the prevalence of inappropriate medication use varies from 16%–82.6% among residents in elderly care homes. It is tempting to speculate that the disparate results in these studies are likely to be due to the different study instruments used.

Two serious drug interactions were identified in our cohort, where alternative medications were recommended. In addition, seven potential adverse drug reactions were also identified that may have necessitated close monitoring of the affected residents. Though the aim of our study was not to monitor the long term adverse effects of poor medication dispensation, some

**Table 5. Medication administration errors among residents.**

| Types of medication administration errors | Frequency (%) [N = 95] * | Examples |
|---|---|---|
| Medication omission | 48 (50.5) | Benzhexol 2 mg twice a day was prescribed, however, the resident did not administer the medication. |
| Wrong dose | 22 (23.2) | Gliclazide 40 mg was administered instead of 80 mg. |
| Wrong time | 15 (15.8) | Atorvastatin 20 mg was administered in the morning instead of night as prescribed. |
| Taken extra doses of medication | 7 (7.3) | Folic acid 1 mg was prescribed once a day, but five residents administered it twice a day. |
| Wrong frequency | 3 (3.2) | Carbidopa+levodopa 275 mg was prescribed four times a day, but the resident administered only morning and night dose. |

*Used as the denominator for calculating percentages

**Table 6. Types of storage errors.**

| Types of storage errors | Frequency (%) | Examples |
|---|---|---|
| | [N = 143] * | |
| Inadequately separated from other medications | 83 (58) | A resident stored all of her antihypertensives, anti-rheumatic drugs and oral hypoglycemics (eight different types of medications) in the same container |
| Use of inappropriate containers | 50 (35) | Glyceryl trinitrate tablets were stored in transparent bottles or plastic bags |
| Suboptimal storage temperature or exposed to sunlight | 10 (7) | Opened insulin vials were stored at room temperature (30°C) |

*Used as the denominator for calculating percentages

of the errors we noted particularly potential adverse drug reactions or serious drug interactions, may have reduced the quality of life of the residents. These adverse findings further justify the necessity to monitor and mandate residents' medication use by trained healthcare professionals.

The mean number of dispensing errors per resident we noted, was considerably higher (15.9 vs. 0.73) compared to a study done by Barber N et al. [5] in UK. This may be because we counted incomplete dispensing label errors, which is a common problem in Sri Lanka [23, 38] and other developing countries [39–41]. Interestingly, previous studies conducted in different settings in Sri Lanka, including hospitals, community pharmacies and household surveys have reported that more than 50% of medicines had incomplete dispensing labels [23, 38]. This is a rather serious predicament particularly in elderly care homes where untrained caregivers are sometimes administering medicines.

One half of residents experienced one or more medication administration errors during the relatively short study period of three months. This is similar to many other studies which report a high incidence of administration errors in elderly care homes [5, 13]. Failing to administer/omitting medications and administering a wrong dose were the most common types of medication errors we observed, a finding similar to care homes in UK [5]. The similarity of these findings between low and high-income countries indicate the universal nature of medication safety issues among this vulnerable group, and the urgent need to address this widespread issue.

The most common storage error found among residents was keeping all medication packets in one container such as a plastic box or a paper bag, and not in their original packaging. Inappropriate storage practices can directly affect the potency of medications. This is especially important for medication with a narrow therapeutic index and could result in loss or reduction in medication efficacy, as well as medication administration errors [42]. However, it should be noted that the majority of Sri Lankans do not use pill organizers or pillboxes to store and organize their medications and they tend to keep their medication in the original envelopes or plastic bags used to dispense the medication. Incidentally, we also observed that some residents did not store opened insulin vials in the refrigerator. Insulin is a protein-containing product which could induce its degradation at high temperature leading to loss of the desired therapeutic effect [43].

Although there have been numerous household surveys conducted worldwide to monitor storage practices of medications at home [24, 31, 44], none have evaluated the storage practices at care homes. Hence, this study, for the first time, demonstrate the necessity and the importance of educating this vulnerable population on appropriate storage practices to safeguard safety, quality and efficacy of the medications they use.

Lack of qualified and trained healthcare staff, busy medication administration rounds and inadequate regular medication reviewing could be the most likely reasons for the high incidence of medication errors among residents in long-term aged care facilities [5]. However, to our knowledge, there is no study that has been conducted in South Asian countries to observe this phenomenon. Our study although limited in sample size provides insights into the medication safety issues related to all three main logistical steps (prescribing, dispensing and medication administration) that entail the medication use process, and hence can be used in a beneficial, proactive way to mitigate these risks among this population.

Strengths and limitations of our study needs to be acknowledged. Although limited to a single district, our study was a multi-center study. However, this district is the commercial capital and the most populated district in Sri Lanka and hence may be generalizable to some extent. Multiple study instruments (prescription and dispensing label review, interview and direct observation) were used to collect data to ensure reliability. In addition, to interviewing and reviewing of medical records, we directly observed administration practices of residents and/or caregivers. Although there is a concern that such observations may affect the prevalence of administration errors, a UK study on observing nurses during medication administration rounds showed that direct observation did not significantly affect the medication administration errors [5]. One major limitation of this study is the limited number of participants we were compelled to recruit and the resultant convenient sample of ours. Although 366 residents were initially approached, most had to be excluded due to the perceived barriers on obtaining reliable information. Therefore, residents with poor cognition, or communication issues, and those without the support of a competent caregiver could not be included. As we failed to document reasons for exclusion, and the demographic characteristics of the excluded residents, post-analysis of excluded residents was also not feasible. Hence, our results should be interpreted with caution due to the foregoing selection bias. Furthermore, as this was a cross-sectional study, we did not monitor patient harm beyond the point of data collection, for instance it was not possible to monitor patient's blood pressure and glucose concentration as these were missing on some of the medical notes. Therefore, the classification of harm caused by prescribing errors could not be classified beyond Category C (an error occurred that reached the patient but did not cause patient harm) and D (an error occurred that reached the patient and required monitoring to confirm that it resulted in no harm to the patient and/or required intervention to preclude harm) of the NCC-MERP classification. Although most residents received support from the caregiver for administering medicines, we did not perform a sub-analysis based on their education level and assistance received by the caregiver, again due to the small sample size. However, we believe that these limitations did not affect the main message of this study, that prescribing, dispensing, medication administration and storage errors are rather rampant among residents of elderly care homes in Sri Lanka irrespective of the education level or supportive care. Other studies have shown that under such circumstances, interventions by competent healthcare personnel can improve medication safety among residents in care homes [25, 45, 46]. Jordan et al in 2015 [46] for instance, conducted a stepped wedge cluster randomized trial to assess the effectiveness of a nurse-led medicines' monitoring intervention for patients with dementia in care homes and reported an improvement in safety of care. Similar interventional studies could be initiated in Sri Lanka to assess effective ways to utilize services of healthcare professionals such as pharmacists, to improve medication safety among residents in elderly care homes.

## Conclusions

The findings of this study indicate that prescribing, dispensing, medication administration and storage errors are prevalent among residents in elderly care homes in Sri Lanka. Wrong administration frequencies in prescriptions, as well as missing information on duration of treatment on dispensing labels, and failing to administer medication, were commonly observed. All medication errors observed were preventable with suitable system changes. As it is immoral and unethical to neglect older adults who reside in elderly care homes we contend these must be considered a priority in healthcare delivery systems. The services of trained resident healthcare professionals are vital to ensure medication safety among this older population. Finally, it is essential that a regular review of the residents' prescriptions is completed by either a doctor or a pharmacist in order to minimize prescribing errors.

## Supporting information

**S1 Table. Definitions and criteria used in the identification of medication errors.**
(DOCX)

**S2 Table. National Coordinating Council for Medication Error Reporting and Prevention (NCC MERP) Index for categorizing medication errors used to categorize prescribing errors.**
(DOCX)

**S1 File. Questionnaire on basic information and demographics of residents.**
(DOCX)

**S2 File. Checklist to assess the appropriateness of dispensing.**
(DOCX)

**S3 File. Checklist to assess the appropriateness of medicine administration.**
(DOCX)

**S4 File. Checklist to assess the appropriateness of medicine storage.**
(DOCX)

**S1 Data.**
(DOCX)

## Acknowledgments

We thank all the heads of the long-term aged care facilities and all the residents who participated in this study. We should extend our sincere gratitude to Professor Lakshman Samaranayake, Professor Emeritus, University of Hong Kong, Hong Kong, and Ms Cathy Lynch, a Clinical Pharmacist and a member of Collaborative of Australian Sri Lankan Pharmacy Practice Education Research (CASPPER) for assisting to improve the scientific content of this manuscript and for assisting in language editing.

## Author Contributions

**Conceptualization:** L. G. T. Shanika, N. R. Samaranayake.

**Data curation:** S. M. S. Prasanna, T. S. B. Cader.

**Formal analysis:** S. M. S. Prasanna, T. S. B. Cader, S. Sabalingam, L. G. T. Shanika, N. R. Samaranayake.

**Methodology:** L. G. T. Shanika, N. R. Samaranayake.

**Supervision:** L. G. T. Shanika, N. R. Samaranayake.

**Writing – original draft:** L. G. T. Shanika.

**Writing – review & editing:** N. R. Samaranayake.

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
