## [Decision Letter · Decision Letter 0]

6 Aug 2019

PONE-D-19-17553

Are medicines used safely among residents in elderly care homes? – An observational multi-center study in Sri Lanka

PLOS ONE

Dear Dr. Samaranayake,

Thank you for submitting your manuscript to PLOS ONE. After careful consideration, we feel that it has merit but does not fully meet PLOS ONE’s publication criteria as it currently stands. Therefore, we invite you to submit a revised version of the manuscript that addresses the points raised during the review process.

We would appreciate receiving your revised manuscript by Sep 20 2019 11:59PM. To enhance the reproducibility of your results, we recommend that if applicable you deposit your laboratory protocols in protocols.io, where a protocol can be assigned its own identifier (DOI) such that it can be cited independently in the future. For instructions see: http://journals.plos.org/plosone/s/submission-guidelines#loc-laboratory-protocols

We look forward to receiving your revised manuscript.

Kind regards,

Prof, Mojtaba Vaismoradi, PhD, MScN, BScN

Academic Editor

PLOS ONE

Journal Requirements:

2. Please include additional information regarding the survey or questionnaire used in the study and ensure that you have provided sufficient details that others could replicate the analyses. For instance, if you developed a questionnaire as part of this study and it is not under a copyright more restrictive than CC-BY, please include a copy, in both the original language and English, as Supporting Information. Moreover, please include more details on how the questionnaire was pre-tested, and whether it was validated.

Reviewers' comments:

Reviewer #1: General

The text requires editing by an English native person

Introduction

The introduction is very weak and should be improved.

The first paragraph is very extensive and should be summarized.

Your statement in the first paragraph is not supported by the references

A review of the literature is too flawed!

Should justify why this research do

Methods

Add the research time to the methods

Number of samples based on what study was calculated?

You have sampled about a quarter of the calculated sample size, which reduces the credibility of the study!

Your study plan (assessment of prescribing, dispensing, medicines administration and storage errors) should be supported by the previous same research.

The sentence: "The Australian Pharmaceutical Formulary (AFP) was used as a reference to assess the appropriateness of dispensing practices" need a suitable reference.

Exactly explain the checklist and questionnaire and its validity and reliability should be mentioned (if done)

In the data analysis section, p <0.05 was considered as a significant level, but in the findings, you did not report p (Although, in my opinion, you only used descriptive statistics).

Line 155: change to SPSS version 16

Because drug errors also are related to caregivers, their information should be taken into account for more comprehensive examination.

The inferential statistics should also be used to analyze the data.

Results

Table 1: Abbreviations should be explained

Tables can be better designed

Other chronic diseases can be added if the information is available

Discussion

You did not provide any information about the health care workers in results, therefore, you should not focus on this issue in the discussion (first paragraph).

Some of the content in the first paragraphs of the discussion is suitable for introduction

In the discussion section, the results should be explained and not repeated (especially in the third paragraph)

In discussion use fewer numbers

Again, you just repeat the results in discussion

The literature review should be more comprehensive and the discussion should be better written

Best regard and good work

Reviewer #2: Thank you for the opportunity to review this interview study. Medication errors remain an important topic, and the prevalence of the problems identified are of concern.

Style

Extensive copy editing will be needed. The manuscript should be revised by a native English speaker.

Abstract

Numbers should be given, alongside %s.

Literature

The reference lists needs to be updated: the more recent citation was 2015, and most citations were pre-2010. There are a plethora of studies and reports in this field, and a more detailed examination of the literature in needed to justify inclusion in Plos One.

Methods

I did not see the dates of data collection.

The types of errors observed should all be defined, with references.

The authors should justify the exclusion of residents with poor cognition and communication. How many potential participants were excluded for these reasons? This introduces a selection bias.

Sample size is not based on any stated outcome.

The instruments used should be referenced or appended.

An old version of spss was used. Proportions can be compared in spss, but I did not see any such comparisons in the results.

Many common long-term conditions do not appear to be included e.g. mental health problems, respiratory conditions, and epilepsy. This limits the generalisation of findings.

Results

A flow chart is needed for residents, in addition to that for homes.

Please explain abbreviations and categories.

This is a descriptive, rather than a quantitative study. This is fine, but the descriptions need to enriched.

The tables all need to be augmented with detailed examples and case reports. To understand the data, readers need to know what was administered and what should have been administered, and how often, with references. Did the potential ADRs cause harm? What happened to the residents? What would have been ‘justifiable indications’ for vitamins etc? (Some authors recommend universal vitamin D.) What were the effects of the interactions? What alternatives were substituted? Were some duplicate medicines added for good reasons? All these prohibitions need references. Which medicines were inappropriately stored? SmPCs should be referenced here.

Discussion

The observational nature of this work is an important limitation. Were the observed vulnerable to any biases? Patient selection further limits generatlisation.

Conclusion

In my view, this study confirms the pervasiveness of the problems, and intervention studies are now needed (Jordan et al 2015).

Jordan et al 2015 Nurse-Led Medicines' Monitoring for Patients with Dementia in Care Homes: A Pragmatic Cohort Stepped Wedge Cluster Randomised Trial

http://dx.plos.org/10.1371/journal.pone.0140203

---

## [Author Response · Author response to Decision Letter 0]

4 Oct 2019

Editor comments

Comment 1: When submitting your revision, we need you to address these additional requirements.

Response 1: Thank you and we have adhered to the journal guidelines to the best of our ability.

Comment 2: Please include additional information regarding the survey or questionnaire used in the study and ensure that you have provided sufficient details that others could replicate the analyses. For instance, if you developed a questionnaire as part of this study and it is not under a copyright more restrictive than CC-BY, please include a copy, in both the original language and English, as Supporting Information. Moreover, please include more details on how the questionnaire was pre-tested, and whether it was validated.

Response 2: We have added the checklists and questionnaires as supporting information.

Comment 3: We suggest you thoroughly copyedit your manuscript for language usage, spelling, and grammar. If you do not know anyone who can help you do this, you may wish to consider employing a professional scientific editing service. 

Response 3: Thank you and we have now subjected the manuscript for language edition. 

Comment 4: We note that you have indicated that data from this study are available upon request. PLOS only allows data to be available upon request if there are legal or ethical restrictions on sharing data publicly. For information on unacceptable data access restrictions, please see http://journals.plos.org/plosone/s/data-availability#loc-unacceptable-data-access-restrictions.

Response 4: Thank you and the data set will be made available as requested if accepted for publication. 

Reviewer comments

Reviewer 1:

Comment 1: The text requires editing by an English native person

Response 1: Thank for the request and the paper was subjected to the edition by a native English speaker. 

Name: Ms Cathy Lynch - a clinical pharmacist and a member of Collaborative of Australian Sri Lankan Pharmacy Practice Education Research (CASPPER)

Introduction

Comment 2: The introduction is very weak and should be improved. The first paragraph is very extensive and should be summarized. Your statement in the first paragraph is not supported by the references. A review of the literature is too flawed! Should justify why this research do.

Response 2: Thank you for the comment. We have now revisited the introduction and improved it. We have also rearranged the introduction section highlighting the importance of conducting this research in Sri Lanka. References have been added where necessary.

Methods

Comment 3: Add the research time to the methods

Response 3: The period was added and shown in line number 163

Comment 4: Number of samples based on what study was calculated?

Response 4: The question is not clear. However, if the query is if we based the sample size calculation on a previous study, then the response is that we used an online sample size calculator and calculated the sample size using stated parameters. This method, however, has been used by others and we added this in the manuscript with appropriate referencing. 

(Line number 109-112) 

Comment 5: You have sampled about a quarter of the calculated sample size, which reduces the credibility of the study!

Response 5: Thank you for the comment and this reduced sample size is one of the limitations in our study. We have acknowledged it in the limitation section (Line number 120 and 336). The study was mainly conducted in the Colombo district (the Commercial capital of Sri Lanka) as it has the highest population density in Sri Lanka. As explained in the methodology, though the records of the Social Services Department shows the existence of 34 elderly care homes, only 14 actually existed and 09 were accessible. We approached each participant in all 9 care homes, but a considerable number did not fulfil the inclusion criteria. Unfortunately, we have not documented the demographics of those excluded nor the specific reason for exclusion, hence difficult to present a systematic flowchart of participant recruitment. This too has now been acknowledged in the limitations. However, we have tried to the best of our ability to showcase the details of participant recruitment and Table 1 which shows how many participants resided in each care home and the % selected for the study has been added.

Comment 6: Your study plan (assessment of prescribing, dispensing, medicines administration and storage errors) should be supported by the previous same research.

Thank you and we agree. Our group has been engaged in research on medication safety in the past and the assessment methodologies of prescribing and dispensing errors are based on our previous experiences which have already been published. Our previous studies and other relevant studies have been added as references (Line number 129-132, 136-138, 143-144). Assessment of drug administration errors (Line number 152-154) and medicine storage errors (Line number 159) are very commonly assessed using direct observation. We have added references to support this methodology. 

Comment 7: The sentence: "The Australian Pharmaceutical Formulary (AFP) was used as a reference to assess the appropriateness of dispensing practices" need a suitable reference.

Response 7: The Australian Pharmaceutical Formulary (AFP) is an accepted reference used to guide pharmacists in good dispensing practices. We have now added the edition number and name of the full reference in the reference list (Line number 147-148). This book has instructions for dispensing labels and the relevant medication administration instructions for patients. This is the National formulary used by pharmacists in Australia.

Comment 8: Exactly explain the checklist and questionnaire and its validity and reliability should be mentioned (if done)

Response 8: Thank you for the comment. All the checklists and questionnaires used have been added as supplementary material. In addition, the questionnaire and checklists were content validated by two other academics pharmacists and were piloted for feasibility among ten elderly persons who were not part of the study. This too has been added to the methodology now (Line number 165-167).

Comment 9: In the data analysis section, p <0.05 was considered as a significant level, but in the findings, you did not report p (Although, in my opinion, you only used descriptive statistics).

Response 9: Thank you for your concern. Yes as we considered only descriptive statistics which we believe is adequate to be inline with our objectives. We did not perform any comparisons and we did not use “P” values. Now we have removed this redundant sentence from the analysis section. 

Comment 10: Line 155: change to SPSS version 16

Response 10: Changed to “Version 16” (Line number 174).

Comment 11: Because drug errors also are related to caregivers, their information should be taken into account for more comprehensive examination.

Response 11: We agree with your comment and this information would have greatly improved our paper. However, the association or relationship between knowledge, education level or other contributory factors of residents or caregivers and the occurrence of medication errors was not an objective of this paper. Besides, a sub-analysis was discouraged due to the already small sample size. We have acknowledged this as a limitation in the manuscript (Line number 124). 

Comment 12: The inferential statistics should also be used to analyze the data.

Response 12: Thank you for your comment. The main objectives of this study were to assess and report the existence and prevalence of medication errors (prescribing, dispensing, administration and storage errors) among this vulnerable population in the South Asian part of the world where such data is scarce. Hence inferential statistics were not used. We could have compared data of each elderly home but did not do so as a sub-analysis would not be of value due to the small sample size. We hope this is acceptable to the reviewers and would be grateful for further guidance. 

Results

Comment 13: Table 1: Abbreviations should be explained

Response 13: Thank you and we have added a list of abbreviations now.

Comment 14: Tables can be better designed

Response 14: Now we have expanded the tables by including examples for each error category (Table 3,4,5, and 6)

Comment 15: Other chronic diseases can be added if the information is available

Response 15: Thank you and the major diseases have been included now. The minority has been added as ‘Others’ with the breakdown in a footnote. Further, on scrutiny, we have detected an error where we have added frequencies instead of percentages. We have corrected this mistake (Table 2).

Discussion

Comment 16: You did not provide any information about the health care workers in results, therefore, you should not focus on this issue in the discussion (first paragraph).

Thank you and we agree. We have now removed this section from the discussion.

Comment 17: Some of the content in the first paragraphs of the discussion is suitable for introduction

In the discussion section, the results should be explained and not repeated (especially in the third paragraph)

In discussion use fewer numbers

Again, you just repeat the results in discussion.

Response 17: Thank you for this constructive comment and now we have rewritten the discussion. Repetition of the results in the discussion has been removed and discussed further with examples from the literature. 

Comment 18: The literature review should be more comprehensive and the discussion should be better written

Response 18: Thank you for the comment and now we have rewritten the discussion considering all your valuable comments.

Reviewer 2:

Comment 1: Style

Extensive copy editing will be needed. The manuscript should be revised by a native English speaker.

Response1: Thank for the request and the paper was edited by a native English speaker.

Abstract

Comment 2: Numbers should be given, alongside %s.

Response 2: Percentages were added (Line numbers 42,43,46 and 48).

Literature

Comment 3: The reference lists needs to be updated: the more recent citation was 2015, and most citations were pre-2010. There are a plethora of studies and reports in this field, and a more detailed examination of the literature in needed to justify inclusion in Plos One.

Response 3: Thank you for the comment and now we have arranged the introduction giving the most recent literature.

Methods

Comment 4: I did not see the dates of data collection.

Response 4: Dates of data collection were added (Line number 163).

Comment 5: The types of errors observed should all be defined, with references.

Response 5: The types of errors observed were based on a list of definitions prepared in-house but based on published literature. The list of definitions have now been added as supporting material (S1 Table) and due references have been added. 

Comment 6: The authors should justify the exclusion of residents with poor cognition and communication. How many potential participants were excluded for these reasons? This introduces a selection bias. 

Response 6: We have now re-worded the exclusion criteria as ‘Residents with poor cognition and communication issues without the assistance of competent caregiver’ as this was the practice we adhered to. 6.9% of the patients included in the study had mental illnesses as they had competent caregivers. Those without caregiver assistance were excluded as they may not be able to provide reliable answers to a medication history interview and may not be able to comprehend the questions asked by researchers. The assistance of caregivers who were informed or competent enough to provide information was not uniformly available at all homes. Hence, we were compelled to exclude them from the study. Due justification has been added to the manuscript now (Line number 338-340).

Comment 7: Sample size is not based on any stated outcome. 

Response 7: The sample size was not based on any primary outcomes as we were unable to find out the baseline statistical figures from the South Asian region which has a similar setup to Sri Lanka. Therefore, it was decided to calculate the sample size based on the prevalence of population aged 60 years and above in Sri Lanka (12.4%, 2016) with a 95% confidence level and 5% of margin of error. This explanation is now added to the manuscript. This method too is supported by a published study by Hasan et al, who has evaluated the medication appropriateness and frailty among residents of aged care homes in Malaysia which is a Southeast Asian country. This reference has been added (Line number 109-112).

Comment 8. The instruments used should be referenced or appended.

Response 8: Thank you for the comment. We used a prescription and medicine label review for detecting prescribing (Line number 136-138) and dispensing errors (Line number 143-144) and direct observation for detecting drug administration (Line number 152-154) and storage errors (Line number 159). The data collection forms and checklists used for gathering this information are now submitted as supporting material (S1 – S4 files). 

Comment 9:. An old version of spss was used. Proportions can be compared in spss, but I did not see any such comparisons in the results.

Response 9: We agree. There was no comparison of proportions. Hence the said sentence has been removed.

Comment 10: Many common long-term conditions do not appear to be included e.g. mental health problems, respiratory conditions, and epilepsy. This limits the generalisation of findings.

Response 10: The list of all the long-term conditions is now incorporated under Table 2. Our sample has 6.9% of residents with mental health and 4.6% of residents with respiratory diseases (Table 2).

Results

Comment 11: A flow chart is needed for residents, in addition to that for homes.

Response 11: Thank you and as previously mentioned in a comment by reviewer 1, we have included a table (Table 1) in the methodology section showing how many residents were living in the care homes and the percentage of residents selected for the study. Those excluded did not fulfil the inclusion criteria. Unfortunately, we have not documented the details of exclusion, hence difficult to present a systematic flowchart of participant recruitment. This too has now been acknowledged in the limitations (Line number 340-342). 

Comment 12: Please explain abbreviations and categories.

Response 12: Abbreviations and categories have been explained.

Comment 13: This is a descriptive, rather than a quantitative study. This is fine, but the descriptions need to enriched.

Response 13: We have used quantitative and qualitative methodologies but the outcomes are presentable numerically. We have included examples where ever possible and tried to enrich the descriptions to the best of our ability. Please guide us further if more is needed.

Comment 14: The tables all need to be augmented with detailed examples and case reports. To understand the data, readers need to know what was administered and what should have been administered, and how often, with references.

Response 14: Thank you and we agree. As previously mentioned in a comment by reviewer 1, we have re-visited the examples given for each medication error and included where possible what should have been given and how often. The references used were BNF and AMH and they have been referenced accordingly. 

Comment 15: Did the potential ADRs cause harm? What happened to the residents? What were the effects of the interactions?

Response 15: None of the residents receiving these inappropriate medicines were harmed at the time of the study and hence belonged to Category C (an error occurred that reached the patient but did not cause patient harm) or Category D (an error occurred that reached the patient and required monitoring to confirm that it resulted in no harm to the patient and/or required intervention to preclude harm) of the NCC MERP Index for Categorizing Medication Errors. Residents were not monitored beyond the point of data collection, hence could not conclude if these prescribing errors resulted in hospitalization or death in a future date. This has been added in the results section now. We have also mentioned this as a limitation of the study now (Line number 342-343).

Comment 16: What would have been ‘justifiable indications’ for vitamins etc? (Some authors recommend universal vitamin D.) 

Response 16: Thank you for the comment. In this study protocol, we were compelled to use a limited number of references to ensure straightforward decision making. In Sri Lanka, the most common reference used by prescribers is the BNF. Hence BNF was used as the primary reference source when detecting prescribing errors. A second reference was also added to confirm the prescribing error. The researchers reviewed the prescription and matched the diagnosis/medical history with medicines prescribed. If the prescribed medicine was not indicated for the disease in at least one of the references used, it was taken as an error. The same method was used for vitamins etc. References used have been added to the reference list now. However, at present the necessity of routine vitamin D supplementation has not been established in our country nor added in the formularies hence was not considered as a requirement for patients. 

Comment 17: What alternatives were substituted? Were some duplicate medicines added for good reasons? All these prohibitions need references. 

Response 17: Duplicate medicines were defined as medicines from the same pharmacological group with the same mechanism of action. Hence duplicate medicines cannot/should not be added for a good reason. Some medicine combinations were not from the same group but have no clinical benefit when used together. These combinations were termed as ‘Inappropriate medicine combinations’. The definition list has now been appended (S1 Table). 

Comment 18: Which medicines were inappropriately stored? SmPCs should be referenced here.

Response 18: We have now added the examples for all of these error types and their categories in Table 6. The references used to guide us on detecting storage errors has also been added in the reference list. 

Discussion

Comment 19: The observational nature of this work is an important limitation. Were the observed vulnerable to any biases? Patient selection further limits generatlisation.

Response 19: Though there is a concern that observation may affect the prevalence of administration error, it is evident from the literature that direct observation had no significant effect on medicine administration errors. This has been added into the discussion now with an appropriate reference (Line number 333-335).

Conclusion

Comment 20: In my view, this study confirms the pervasiveness of the problems, and intervention studies are now needed (Jordan et al 2015).

Jordan et al 2015 Nurse-Led Medicines' Monitoring for Patients with Dementia in Care Homes: A Pragmatic Cohort Stepped Wedge Cluster Randomised Trial

http://dx.plos.org/10.1371/journal.pone.0140203

Response 20: Thank you and we agree. We have added this phrase to the discussion with this reference (Line number 354-358).

---

## [Decision Letter · Decision Letter 1]

30 Oct 2019

PONE-D-19-17553R1

Are medicines used safely among residents in elderly care homes? – An observational multi-centred study in Sri Lanka

PLOS ONE

Dear Dr. Samaranayake,

Thank you for submitting your manuscript to PLOS ONE. After careful consideration, we feel that it has merit but does not fully meet PLOS ONE’s publication criteria as it currently stands. Therefore, we invite you to submit a revised version of the manuscript that addresses the points raised during the review process.

We would appreciate receiving your revised manuscript by Dec 14 2019 11:59PM. To enhance the reproducibility of your results, we recommend that if applicable you deposit your laboratory protocols in protocols.io, where a protocol can be assigned its own identifier (DOI) such that it can be cited independently in the future. For instructions see: http://journals.plos.org/plosone/s/submission-guidelines#loc-laboratory-protocols

We look forward to receiving your revised manuscript.

Kind regards,

Prof, Mojtaba Vaismoradi, PhD, MScN, BScN

Academic Editor

PLOS ONE

Reviewers' comments

Reviewer #1: The authors have addressed most of my concerns regarding the original version of the paper and made substantial improvements. However, there are still some minor revisions that I believe are necessary prior to publication. Specifically:

General

The entire manuscript should be reviewed in term of language punctuation

Introduction

“Labour” change to ” labor”

“Ageing” change to “aging”

Line 57: are change to “is”

Line 69: ‘carers” should be replaced with a better word

Line 71: United State (US)

Line 90: “may be” change to “maybe”

Line 90: “to” change to “from”

Methods

Line 115: However,

Result

Line 198: Eighty-five percent of residents

1ine 203: most

Discussion

Line 293: “carers” should be replaced with a better word

Line 311: “household”

Reviewer #2: Thank you for the opportunity to re-review this observational study.

Thank you for adding examples of care – this makes it much easier to understand the problems.

Abstract

Line 43, administration error numbers were not normally distributed, so the median and 25th-75th centiles should be quoted. Please could the statistician also check that other parameters were normally distributed before quoting just the means and SDs.

Introduction

Line 67. Please could you confirm that the social care workers were all unpaid? This would not be the case in the UK, so an explanation would be helpful.

Lines 80-83. It seems that there are conclusions in the introduction.

Sample size

This is a convenience sample. The calculation presented bears no relation to the outcomes investigated. I suggest remove the calculation and explain the factors limiting the study size.

Sample selection

Line 14. Did the homes refuse access or consent?

How many residents were excluded because they were <60? How many actually refused?

Process

Line 137. Were these references the current editions at the time of the study? if so, need to state this.

Errors

Line 203. Need to say in what way the frequencies were incorrect.

Discussion

Lines 343 et seq. to state that no harm occurred, the BPs and glucose concentrations need to be given. What checks were undertaken to ensure there were no signs and symptoms that might have attributable to the errors? If none, this might be a limitation of the study.

References

Some updating is needed e.g. annual health bulletin 2016 [2].

Style

There are a few problems with English, which will need the attention of copy editors.

---

## [Author Response · Author response to Decision Letter 1]

20 Feb 2020

Reviewer 1: The authors have addressed most of my concerns regarding the original version of the paper and made substantial improvements. However, there are still some minor revisions that I believe are necessary prior to publication. Specifically:

Comment 1

General

The entire manuscript should be reviewed in term of language punctuation

Response 1 

The manuscript was edited by a native English speaker previously and now being thoroughly reviewed by the authors.

Comment 2

Introduction

“Labour” change to ” labor”

Corrected (Please refer Line 62)

“Ageing” change to “aging”

Corrected (Please refer Line 55)

Line 57: are change to “is” 

Corrected (Please refer Line 56)

Comment 3

Line 69: ‘carers” should be replaced with a better word

Response 3

Replaced with caregivers (Please refer Line 70)

Comment 4

Line 71: United State (US)

Response 4

Changed (Please refer Line 72)

Comment 5

Line 90: “may be” change to “maybe”

Response 5

Changed (Please refer Line 88)

Comment 6

Line 90: “to” change to “from”

Response 6

Changed (Please refer Line 89)

Comment 7

Methods

Line 115: However,

Response 7

Edited (Please refer Line 111)

Comment 8

Result

Line 198: Eighty-five percent of residents

Response 8

Edited (Please refer Line 191)

Comment 9

Line 203: most

Response 9

Changed (Please refer Line 196)

Comment 10

Discussion

Line 293: “carers” should be replaced with a better word

Response 10

Replaced with the word “caregivers” (Please refer Line 288)

Comment 11

Line 311: “household”

Response 11

Edited (Please refer Line 305)

Reviewer 2: 

Abstract

Comment 1: Line 43, administration error numbers were not normally distributed, so the median and 25th-75th centiles should be quoted. Please could the statistician also check that other parameters were normally distributed before quoting just the means and SDs.

Response 1

Now we have shown both the mean (standard deviation) and median (25th-75th centiles) values in the abstract and the main text.

Introduction

Comment 2: Line 67. Please could you confirm that the social care workers were all unpaid? This would not be the case in the UK, so an explanation would be helpful.

Response 2

Some of the social care workers at residential care homes work voluntarily and are not paid a formal salary while some are hired as paid staff. The sentence was re-worded as follows.

“Most of these facilities do not employ trained healthcare professionals but employ staff who have not received any formal training on safe use of medicines.”

Comment 3: Lines 80-83. It seems that there are conclusions in the introduction.

Response 3

Thank you and we agree. Now we have removed this part from the introduction.

Sample size

Comment 4: This is a convenience sample. The calculation presented bears no relation to the outcomes investigated. I suggest remove the calculation and explain the factors limiting the study size.

Response 4

Thank you and we agree. Now we have removed the description of the sample size calculation.

Sample selection

Comment 5: Line 14. Did the homes refuse access or consent?

Response 5

The homes did not grant permission for researchers to approach the residents and thereby refused access. Now we have reworded it in the manuscript (Please refer Line 109).

Comment 6: How many residents were excluded because they were <60? How many actually refused?

Process

Response 6

Unfortunately, we have not documented the demographics of those excluded nor the specific reason for exclusion. We only considered if participants matched the inclusion criteria. Hence, it is not possible to answer this query.

Comment 7: Line 137. Were these references the current editions at the time of the study? if so, need to state this.

Response 7

Yes, the study was conducted in 2016 and the team used the current latest available editions at that time. The following was added,

“The current and latest available editions of British National Formulary (BNF) (Version 71) [26], the Australian Medicines Handbook (AMH) [27] and Medscape Pharmacists [28] were used as references for detecting prescribing errors” (Please refer Line 130).

.

Errors

Comment 8: Line 203. Need to say in what way the frequencies were incorrect.

Response 8

Now we have discussed it (Please refer Line 196).

Discussion 

Comment 9: 

Lines 343 et seq. to state that no harm occurred, the BPs and glucose concentrations need to be given. What checks were undertaken to ensure there were no signs and symptoms that might have attributable to the errors? If none, this might be a limitation of the study.

Response 9

This observational study was carried out through reviewing documents and patient interview. As blood pressure values and glucose concentrations were not recorded on the patients’ medication documents, it was not possible to identify medication related patient harm. This has now been acknowledged under limitation (Please refer Line 334).

References

Comment 10: Some updating is needed e.g. annual health bulletin 2016 [2].

Thank you however, in Sri Lanka, there is no health bulletin publication available after 2016.

Style

Comment 11: There are a few problems with English, which will need the attention of copy editors.

Response 11

The manuscript was edited by a native English speaker previously and now being thoroughly reviewed by the authors.

---

## [Decision Letter · Decision Letter 2]

6 Mar 2020

PONE-D-19-17553R2

Are medicines used safely among residents in elderly care homes? – An observational multi-centred study in Sri Lanka

PLOS ONE

Dear Dr. Samaranayake,

Thank you for submitting your manuscript to PLOS ONE. After careful consideration, we feel that it has merit but does not fully meet PLOS ONE’s publication criteria as it currently stands. Therefore, we invite you to submit a revised version of the manuscript that addresses the points raised during the review process.

We would appreciate receiving your revised manuscript by Apr 20 2020 11:59PM. To enhance the reproducibility of your results, we recommend that if applicable you deposit your laboratory protocols in protocols.io, where a protocol can be assigned its own identifier (DOI) such that it can be cited independently in the future. For instructions see: http://journals.plos.org/plosone/s/submission-guidelines#loc-laboratory-protocols

We look forward to receiving your revised manuscript.

Kind regards,

Prof, Mojtaba Vaismoradi, PhD, MScN, BScN

Academic Editor

PLOS ONE

Reviewers' comments:

Reviewer #1: The quality of the paper considerably improved than the original version. I recommend this submission for acceptance with a minor revision: some lines and arrows are not straight in figure 1 that should be reformatted.

Reviewer #2: Thank you for the opportunity to re-read this paper. Copy editing is needed.

I think it should be explained in the paper that some workers in care homes were unpaid, and worked voluntarily.

The sample size should be based on the primary outcome. This does not appear to be the case. It could be removed, as explained previously.

The next text indicates that 5 of 34 homes refused access, and 9 were recruited. Is there a typographical error? Line 109

Failure to note reason for exclusion is a major limitation, and should be discussed.

The case studies make this an interesting paper on an international topic that should be highlighted.

---

## [Author Response · Author response to Decision Letter 2]

20 Apr 2020

Response to reviewer comments

Reviewer 1: The quality of the paper considerably improved than the original version. I recommend this submission for acceptance with a minor revision: some lines and arrows are not straight in figure 1 that should be reformatted.

Thank you for the comment and now we have done the required changes in the Figure 1.

Reviewer 2: 

Thank you for the opportunity to re-read this paper. 

Comment 1: Copy editing is needed.

Response 1: Thank you and in addition to English editing done by Ms Cathy Lynch, a Clinical Pharmacist and a member of Collaborative of Australian Sri Lankan Pharmacy Practice Education Research (CASPPER) in the previous submission the entire manuscript has been revisited by Professor Lakshman Samaranayake, Professor Emeritus, University of Hong Kong , Hong Kong and copy edited the article thoroughly. We hope this effort is satisfactory. 

Comment 2: I think it should be explained in the paper that some workers in care homes were unpaid, and worked voluntarily.

Response 2: Please refer line 68-70. This comment was already addressed in the previous submission but we have now further emphasized that most workers are unpaid and work on a voluntary basis.

Comment 3: The sample size should be based on the primary outcome. This does not appear to be the case. It could be removed, as explained previously.

Response 3: In your previous comments, you had advised us to remove the sample size calculation and discuss the limitations. Taking this comment into consideration, we had already removed the sample size calculation from the methodology and explained the factors limiting the study sample size. However, taking this comment into consideration again, we have further elaborated that the sample selected for study is a convenience sample. We have also re-worded the relevant paragraph for better clarity (Line 107 – 115). 

Comment 4: The next text indicates that 5 of 34 homes refused access, and 9 were recruited. Is there a typographical error? Line 109

Response 4: It was not a typographical error. Of 34 elderly care homes registered, only 14 were functioning at the time of study. Out of the 14 functioning, 5 refused access and remaining 9 were approached. It is also mentioned in the manuscript text (Line 109 – 115). 

Comment 5: Failure to note reason for exclusion is a major limitation, and should be discussed.

Response 5: We have not recorded the individual reasons of the excluded sample, however, the overall elders who were excluded did not comply with at least one of our inclusion criteria. We have already acknowledged this in the limitation section (Line 342 -348). We have also acknowledged this limitation in the section dealing with sample size (Line 114-115) .

Comment 6: The case studies make this an interesting paper on an international topic that should be highlighted.

Response 6: Thank you for the comment and we believe no changes are required here.

---

## [Decision Letter · Decision Letter 3]

7 May 2020

Are medications safely used by residents in elderly care homes? – A multi-centre observational study from Sri Lanka

PONE-D-19-17553R3

Dear Dr. Samaranayake,

We are pleased to inform you that your manuscript has been judged scientifically suitable for publication and will be formally accepted for publication once it complies with all outstanding technical requirements.

With kind regards,

Prof, Mojtaba Vaismoradi, PhD, MScN, BScN

Academic Editor

PLOS ONE

Reviewers' comments:

Reviewer #1: All comments have been addressed

Reviewer #2: All comments have been addressed

---

## [Editor Report · Acceptance letter]

27 May 2020

PONE-D-19-17553R3 

Are medications safely used by residents in elderly care homes? – A multi-centre observational study from Sri Lanka 

Dear Dr. Samaranayake:

I am pleased to inform you that your manuscript has been deemed suitable for publication in PLOS ONE. Congratulations! Your manuscript is now with our production department. 

With kind regards,

on behalf of

Professor Mojtaba Vaismoradi 

Academic Editor

PLOS ONE